# PASSAGE: Ensuring Completeness and Responsiveness of Public SPARQL Endpoints with SPARQL Continuation Queries

## Abstract

Being able to query online public knowledge graphs such as Wikidata or DBpedia is extremely valuable. However, these queries can be interrupted due to the fair use policies enforced by SPARQL endpoint providers, leading to incomplete results. While these policies help maintain the responsiveness of public SPARQL endpoints, they compromise the completeness of query results, which limits the feasibility of various downstream tasks. Ideally, we should not have to choose between completeness and responsiveness. To address this issue, we introduce and formalize the concept of *SPARQL continuation queries*. When a SPARQL endpoint interrupts a query, it returns partial results along with a SPARQL continuation query to retrieve the remaining results. If the continuation query is also interrupted, the process repeats, generating further continuation queries until the complete results are obtained. In our experimentation, we show that our continuation server PASSAGE ensures completeness and responsiveness while delivering high performance.

### ACM Reference Format:
Anonymous Author(s). 2024. PASSAGE: Ensuring Completeness and Responsiveness of Public SPARQL Endpoints with SPARQL Continuation Queries. In . ACM, New York, NY, USA, 12 pages. https://doi.org/10.1145/nnnnnnn.nnnnnnn

## 1 Introduction

**Context and motivation:** Linked Open Data (LOD) principles have led to the publication of billions of RDF triples [8, 24]. The ability to query online public SPARQL endpoints such as Wikidata or DBpedia is extremely valuable. However, SPARQL queries are often too long or complex, which violates the fair use policy applied by public SPARQL endpoint providers [4]. While these policies are mandatory to ensure service responsiveness, they compromise the completeness of query results, wasting resources to compute incomplete queries.

To illustrate, consider the query Q1 of Figure 1 that retrieves the women leading cities in Europe. This query times out on Wikidata after 60 seconds, returning only partial results. The partial results of Q1 remain useless for answering the query. When completeness is not ensured, many downstream tasks, such as processing aggregate queries [12], creating portals [15], indexing [18], or computing summaries for federation engines [21], cannot be performed.

```
SELECT ?mayor WHERE {
  ?country wdt:P30 wd:Q46.      # countries of Europe |tp1|=30650
  ?city wdt:P17 ?country.       # and their cities    |tp2|=17913461
  ?city wdt:P6 ?mayor.          # whose mayor         |tp3|=32464
  ?mayor wdt:P21 wd:Q6581072 }  # is a woman          |tp4|=2433035
```

**Figure 1: The Query $Q_1$ about women leading cities in Europe times out after 60 seconds on Wikidata.**

**Related works:** Different approaches have been proposed to ensure both completeness and responsiveness: Linked Data Fragments (LDF) [14, 27] and variants [1, 13], web preemption principle [19], Smart-KG [7] or WiseKG [6]. All these approaches deliver different trade-offs in terms of performance but raise two important issues: the execution time of a single query can be seriously impacted compared to current SPARQL engines, and more importantly, they are not compliant with SPARQL endpoints, which is a strong limitation for adoption. The research question is how to provide a SPARQL endpoint able to deliver completeness, responsiveness, and high performance.

**Approach and contributions:** Inspired by the concept of continuations in programming languages and web [22, 25], we introduce the notion of *SPARQL continuation queries*. The idea is simple: when a SPARQL endpoint server reaches its time quota, it returns partial results along with a SPARQL query designed to return missing results, i.e., a continuation query. If the user wants to obtain missing results, she sends the continuation query to the SPARQL server, which may return partial results and another continuation query. The complete results of the original query are obtained by combining the partial results from all continuation queries. To the best of our knowledge, this is the first proposal to combine responsiveness, completeness, and compliance with the SPARQL standard. The contributions of this paper are as follows:

- We introduce and formalize the concept of *continuation queries* and define the continuation query problem.
- We propose PASSAGE as a solution to the continuation query problem. We provide a formal framework that models partial executions and continuous evaluations. We prove its correctness and termination, and we analyze its complexity.
- We developed a SPARQL query engine built on the Blazegraph storage system, capable of executing core SPARQL queries with continuations. The remaining SPARQL operators are executed through the Comunica smart client [26].
- We compare the performance of PASSAGE against state-of-the-art SPARQL engines on the Wikidata benchmark [3]. Experimental results show that PASSAGE achieves performance comparable to Blazegraph and outperforms Apache Jena while ensuring completeness and responsiveness.

This paper is organized as follows: Section 2 defines the continuation query problem. Section 3 presents PASSAGE, our approach

for solving the continuation query problem. Section 4 reviews the state-of-the-art SPARQL query engines that ensure both responsiveness and completeness. Section 5 presents our experimental results. Section 6 concludes and outlines future work.

## 2 The Continuation Query Problem

We assume that the reader is familiar with RDF and core SPARQL [17, 20], i.e. triple patterns, basic graph patterns, joins, unions, filters, and optionals. SPARQL evaluation semantics are detailed in [17, 20]. In short, the evaluation of a SPARQL query $Q$ over a graph $G$ is defined as a function $[\![Q]\!]_G$ which returns a bag of mappings.

The key idea of our approach is quite simple: when a SPARQL endpoint has to interrupt a query, instead of only returning partial results, it also returns a continuation query that can compute missing results.

*Definition 2.1 (Continuation query).* Let $\lfloor\!\lfloor Q\rfloor\!\rfloor_G$ be a partial evaluation function of $Q$ over $G$, i.e., it returns partial results. The SPARQL continuation query $Q_c$ of $\lfloor\!\lfloor Q\rfloor\!\rfloor_G$ returns the missing results of $Q$ over $G$: $[\![Q]\!]_G = \lfloor\!\lfloor Q\rfloor\!\rfloor_G \uplus [\![Q_c]\!]_G$.

Problem 1 (Continuation queries). *As continuation queries might be longer than allowed by time quotas, the problem is: how to compute a finite sequence of continuation queries $Q_c^1, \ldots, Q_c^n$, where $Q_c^{i+1}$ is the continuation query of $Q_c^i$, and $Q_c^1$ the continuation query of $Q$, such that it provides correct and complete results:* $[\![Q]\!]_G = \lfloor\!\lfloor Q\rfloor\!\rfloor_G \underset{1\leq i\leq n}{\uplus} \lfloor\!\lfloor Q_c^i\rfloor\!\rfloor_G$.

## 3 PASSAGE: SPARQL Continuation Queries

PASSAGE overhauls the notion of partial evaluation to include continuation queries that allow retrieving the remaining results of a partial evaluation.

Like the traditional SPARQL evaluation [17, 20], the *continous evaluation* of a query $Q$ over a graph $G$ includes an additional parameter: a bag of mappings $\Omega$ that represents intermediate results (also called environment [17]). Initially, $\Omega$ is a singleton set containing the empty mapping $\mu_\emptyset$ with an empty domain that is compatible with any mapping. Therefore, $(\!|Q|\!)_G^{\mu_\emptyset}$ corresponds to the evaluation of the query without restrictions $[\![Q]\!]_G$.

*Definition 3.1 (Continuous evaluation $(\!|Q|\!)_G^\Omega$).* The *continuous* evaluation of a query $Q$ over a graph $G$ with a bag of mappings $\Omega$ as an environment, denoted $(\!|Q|\!)_G^\Omega$, returns $(\Omega_p, Q_c)$. $\Omega_p$ is a partial query result, i.e., a bag of solution mappings compatible with $\Omega$ ($\Omega_p \subseteq \Omega \bowtie [\![Q]\!]_G$) and $Q_c$ is $Q$'s continuation query, such that $\Omega \bowtie [\![Q]\!]_G = \Omega_p \uplus [\![Q_c]\!]_G$.

### 3.1 Requirement

The rewriting rules that create continuation queries of PASSAGE rely on the assumption that the evaluation of a triple pattern is deterministic and returns a list of mappings: $[\![tp]\!]_G^\mu = [\mu_1, \ldots \mu_{card([\![tp]\!]_G \bowtie \mu)}]$. While this constraint on triple pattern evaluations is stronger than SPARQL's [5], many SPARQL engines such as Blazegraph or Apache Jena rely on data structures such as B-Trees that already return such a list of elements deterministically.

With this assumption, the evaluation of a *Slice* of triple pattern also becomes deterministic and returns a list of mappings. Then, a

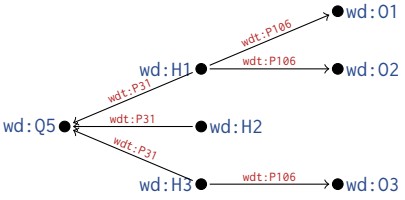

**Figure 2: An example of graph comprising 6 triples about humans (wd:Q5) and occupations (wdt:P106).**

solution to the *continuation queries problem* for triple patterns consists of evaluating disjoint slices of the triple pattern and concatenate their lists of solution mappings as: $[\![tp]\!]_G^\mu = [\![Slice(tp, 0, i)]\!]_G^\mu \cdot [\![Slice(tp, i, card([\![tp]\!]_G \bowtie \mu) - i)]\!]_G^\mu$ where $0 \leq i < card([\![tp]\!]_G \bowtie \mu)$.

### 3.2 Core SPARQL Evaluation

Core SPARQL includes triple patterns, joins, unions, optionals, and filters [20]. However, the specification of the evaluation for SPARQL continuation queries starts with the empty case, allowing simplification of logical plans:

*Definition 3.2 (Empty continuation $(\!|P|\!)_G^\emptyset$ and $(\!|\{\}|\!)_G^\Omega$).* Let $P$ be a graph pattern, and $\Omega$ be a bag of mappings, the evaluation of:

- an empty environment is $(\!|P|\!)_G^\emptyset = (\emptyset, \{\})$;
- an empty graph pattern is $(\!|\{\}|\!)_G^\Omega = (\Omega, \{\})$.

The smallest unit of core SPARQL is the triple pattern, whose evaluation produces mappings. The continuation query requires the OFFSET of $Slice(P, offset, limit)$, which we simplify to $Slice(P, offset)$ when the *limit* equals the remaining the number of results.

*Definition 3.3 (Triple pattern evaluation $(\!|tp|\!)_G^\Omega$).* Let a triple pattern evaluation $[\![tp]\!]_G^\mu = [\mu_1, \ldots \mu_k]$ where $k = card([\![tp]\!]_G \bowtie \mu)$ and i, $0 \leq i \leq k$, a point when the *continuous* evaluation of $tp$ with environment $\mu$ can be stopped. The *continuous* evaluation of a triple pattern $tp$ with an environment $\mu$ is defined as:

$$(\!|tp|\!)_G^\mu = (\{\!|\mu_1, \ldots \mu_i|\!\}, Slice(Extend(\mu, tp), i))$$

$$\text{WHERE } Slice(Extend(\mu, tp), i) = \begin{cases} \{\} & \text{IF } i \geq card([\![tp]\!]_G \bowtie \mu) \\ Extend(\mu, tp) & \text{IF } i = 0 \\ Slice(tp, i) & \text{IF } \mu = \mu_\emptyset \end{cases}$$

*Definition 3.4 (Sliced triple pattern evaluation).* Let a sliced triple pattern evaluation $[\![Slice(Extend(\mu, tp), i)]\!]_G^{\mu'} = [\mu_{i+1}, \ldots \mu_{i+j}, \ldots \mu_n]$ where $n = card(\mu' \bowtie \mu \bowtie [\![tp]\!]_G)$, the *continuous* evaluation of $Slice(Extend(\mu, tp), i)$ with environment $\mu'$ can be stopped at any point $1 \leq j \leq n - i$. That is, for any $j$ between 1 and $n - i$:

$$(\!|Slice(Extend(\mu, tp), i)|\!)_G^{\mu'} = \begin{array}{l}(\{\!|\mu_{i+1}, \ldots \mu_{i+j}|\!\}, \\ Slice(Extend(\mu, tp), i + j))\end{array}$$

*Definition 3.5 (Extend evaluation $(\!|Extend(E, P)|\!)_G^\mu$).* The evaluation of an extend with a mapping $\mu$ over a graph pattern $P$ is:

$$(\!|Extend(\mu, P)|\!)_G^{\mu'} = (\mu \bowtie \Omega_p, Extend(\mu, Q_p)) \text{ WHERE } (\!|P|\!)_G^{\mu'} = (\Omega_p, Q_p)$$

$$(\!|Extend(\mu)|\!)_G^{\mu'} = (\!|Extend(\mu, \{\})|\!)_G^{\mu'}$$

*Example 3.6.* To illustrate, let us consider the Graph $G_{ex}$ of Figure 2 that contains 6 triples, 3 triples of which are about humans: wd:H1, wd:H2, and wd:H3. An end-user wants to enumerate all human beings in the graph with the SPARQL query:

```
SELECT * WHERE { ?human wdt:P31 wd:Q5 } # get all human beings
```

The *continuous* evaluation of this query $(\!|tp|\!)_G^{\mu_\emptyset}$ where $tp = (\textit{?human}$ wdt:P31 wd:Q5) may be interrupted at any point between 1 and $card(tp) = 3$. For example, being interrupted after the first mapping we obtain:

$$(\!|tp|\!)_G^{\mu_\emptyset} = (\{\!|\{\textit{?human} \rightarrow \text{wd:H1}\}|\!\}, Slice(Extend(\mu_\emptyset, tp), 1))$$
$$= (\{\!|\{\textit{?human} \rightarrow \text{wd:H1}\}|\!\}, Slice(tp, 1))$$

The end-user receives a first partial result $\{\textit{?human} \rightarrow \text{wd:H1}\}$, along with the continuation query that corresponds to the following SPARQL query:

```
SELECT * WHERE { ?human wdt:P31 wd:Q5 } OFFSET 1 # skip first result
```

To retrieve missing results, the end-user sends back this regular SPARQL query. It is designed to skip the first produced result thanks to the OFFSET clause. Again, the continuous evaluation stops after having computed one mapping, resulting in:

$$(\!|Slice(tp, 1)|\!)_G^{\mu_\emptyset} = (\{\!|\{\textit{?human} \rightarrow \text{wd:H2}\}|\!\}, Slice(tp, 2))$$

The end-user receives a second partial result $\{\textit{?human} \rightarrow \text{wd:H2}\}$ with a SPARQL continuation query corresponding to:

```
SELECT * WHERE { ?human wdt:P31 wd:Q5 } OFFSET 2 # skip two results
```

The end-user sends back this SPARQL query designed to skip the two first results. When the continuous evaluation stops again after having produced one mapping, the result is:

$$(\!|Slice(tp, 1)|\!)_G^{\mu_\emptyset} = (\{\!|\{\textit{?human} \rightarrow \text{wd:H3}\}|\!\}, Slice(tp, 3))$$
$$= (\{\!|\{\textit{?human} \rightarrow \text{wd:H3}\}|\!\}, \{\})$$

The continuation query becomes the empty graph pattern, since the offset reached the cardinality of the triple pattern. When the end-user receives the third result, she acknowledges that she got correct and complete results since the last continuation query is empty.

The solution of the continuation queries problem has not been computed beforehand, but it is obtained from the partial executions completed within the allowed time quota. For this example, the sequence of continuation queries is $Slice(tp, 1)$, $Slice(tp, 2)$, $\{\}$. The partial evaluation of these queries provides the two solutions that complete the partial evaluation of $tp$.

Core SPARQL includes conjunctive queries with the join operator between two graph patterns. Since the results of one side depends on the results of the other side, we devise a general rule for environments with multiple mappings as the union of the evaluations over every mapping:

$$(\!|P|\!)_G^\Omega = (\Omega_h \uplus \Omega_t, Union(Q_h, Q_t)) \text{ WHERE } \begin{cases} \Omega = \mu \uplus \Omega_o \\ (\!|P|\!)_G^\mu = (\Omega_h, Q_h) \\ (\!|P|\!)_G^{\Omega_o} = (\Omega_t, Q_t) \end{cases}$$

The evaluation of a join operator can be stopped at any point during the evaluation of its operands. For clarity, let us assume that the left operand is evaluated first, with the resulting mappings passed as the environment to the right operand. Both operands could be stopped. In this case, the continuation query of a join operator includes the remaining evaluation of its left operand (which still needs to be operated with the right operand ) and the remaining evaluation of the right operand with the new environment.

*Definition 3.7 (Join evaluation $(\!|Join(P_l, P_r)|\!)_G^\mu$).* The *continuous* evaluation of a join between two graph patterns $P_l$ and $P_r$ with environment $\mu$ is:

$$(\!|Join(P_l, P_r)|\!)_G^\mu = (\Omega_{lr}, Q_c) \text{ WHERE } \begin{cases} (\!|P_l|\!)_G^\mu = (\Omega_l, Q_l) \\ (\!|P_r|\!)_G^{\Omega_l} = (\Omega_{lr}, Q_{lr}) \\ Q_c = Union(Join(Q_l, P_r), Q_{lr}) \end{cases}$$

The SPARQL continuation query of a conjunctive query includes a union, therefore, we define the continuous evaluation of disjunctions. The continuation query of a union operator unions solution mappings and continuation queries of both operands:

*Definition 3.8 (Union evaluation $(\!|Union(P_l, P_r)|\!)_G^\mu$).* The *continuous* evaluation of a union between two graph patterns $P_l$ and $P_r$ with a mapping $\mu$ as environment is:

$$(\!|Union(P_l, P_r)|\!)_G^\mu = (\Omega_l \uplus \Omega_r, Q_c) \text{ WHERE } \begin{cases} (\!|P_l|\!)_G^\mu = (\Omega_l, Q_l) \\ (\!|P_r|\!)_G^\mu = (\Omega_r, Q_r) \\ Q_c = Union(Q_l, Q_r) \end{cases}$$

We consider the evaluation of basic graph patterns (BGPs) as an instance of a join evaluation where $P_l$ is the first triple pattern and $P_r$ corresponds to the remaining triple patterns.

*Example 3.9.* To illustrate the evaluation of conjunctive queries, let us consider the Graph $G_{ex}$ of Figure 2. An end-user wants to retrieve all human beings along with their occupations – a query $Q_{bgp}$ that times out after 60 seconds on the Wikidata public SPARQL endpoint:

```
SELECT * WHERE {
  ?human wdt:P31 wd:Q5 .        # tp1: all human beings
  ?human wdt:P106 ?occupation } # tp2: along with their occupations
```

The continuous evaluation of the query $Join(tp_1, tp_2)$ where $tp_1 = (\textit{?human}\,\text{wdt:P31 wd:Q5})$ and $tp_2 = (\textit{?human}\,\text{wdt:P106 \textit{?occupation}})$ is $(\!|Join(tp_1, tp_2)|\!)_G^{\mu_\emptyset}$. Assuming an interruption after having read the first mapping of $tp_1$, the resulting continuous evaluation is:

$$(\!|Join(tp_1, tp_2)|\!)_G^{\mu_\emptyset} = \begin{array}{l} (\emptyset, Union(Join(Slice(tp_1), 1), tp_2), \\ \qquad\qquad Extend(\{\textit{?human} \rightarrow \text{wd:H1}\}, tp_2)) \end{array}$$

The end-user receives no results yet, with a SPARQL continuation query corresponding to:

```
SELECT * WHERE {
  { SELECT * WHERE { # part I
    BIND ( wd:H1 AS ?human ) # skip reading the first triple
    ?human wdt:P106 ?occupation }
  } UNION {
    { SELECT * WHERE { ?human wdt:P31 wd:Q5 } OFFSET 1 } # part IIA
    ?human wdt:P106 ?occupation }} # part IIB
```

The end-user sends back this query $Q_{bgp}^1$ to the PASSAGE endpoint to get complete results. Again, assuming an interruption after having read the first mapping of $tp_2$, the result of the continuous evaluation is the following:

$$(\!|Q_{bgp}^1|\!)_G^{\mu_\emptyset} = \begin{array}{l} (\{\!|\{\textit{?human} \rightarrow \text{wd:H1}, \textit{?occupation} \rightarrow \text{wd:O1}\}|\!\}, \\ Union(Join(Slice(tp_1, 1), tp_2), \\ \qquad Extend(\{\textit{?human} \rightarrow \text{wd:H1}\}, Slice(tp_2, 1)))) \end{array}$$

The resulting continuation query $Q^2_{bgp}$ is mostly similar to $Q^1_{bgp}$ since only the left graph pattern of the union progressed; and this graph pattern is already a sliced triple pattern. The end-user receives a result, and a SPARQL continuation query corresponding to:

```
SELECT * WHERE {
  { SELECT * WHERE { # part I
    BIND ( wd:H1 AS ?human )
    ?human wdt:P106 ?occupation } OFFSET 1 # skip the first result
  } UNION {
    { SELECT * WHERE { ?human wdt:P31 wd:Q5 } OFFSET 1 } # part IIA
    ?human wdt:P106 ?occupation }} # part IIB
```

The end-user sends back this continuation query $Q^2_{bgp}$ to the PAS-SAGE endpoint to get complete results. This time, the endpoint manages to fully compute the query $(\!(Q^2_{bgp})\!)^{\mu_\emptyset}_G = ([\![Q^2_{bgp}]\!]^{\mu_\emptyset}_G, \{\})$, i.e., the end-user receives two additional results $\{?human \rightarrow$ wd:H1, $?occupation \rightarrow$ wd:O2$\}$ and $\{?human \rightarrow$ wd:H3, $?occupation \rightarrow$ wd:O3$\}$; and acknowledges query termination with the empty continuation query.

*Definition 3.10 (Optional evaluation $(\!(LeftJoin(P_l, P_r))\!)^\mu_G)$.* The evaluation of a left join (or optional) between two graph patterns $P_l$ and $P_r$ with a mapping $\mu$ as environment is:

$$(\!(LeftJoin(\{\}, P_r))\!)^\mu_G = \begin{cases} (\{\!| \mu |\!\}, \{\}) & \text{IF } (\!(P_r)\!)^\mu_G = (\emptyset, \{\}) \\ (\{\!| \mu |\!\} \Join \Omega_r, \{\}) & \text{IF } (\!(P_r)\!)^\mu_G = (\Omega_r, \{\}) \\ (\emptyset, LeftJoin(Extend(\mu), Q_r)) & \text{IF } (\!(P_r)\!)^\mu_G = (\emptyset, Q_r) \\ (\!(P_r)\!)^\mu_G & \text{OTHERWISE} \end{cases}$$

$$(\!(LeftJoin(P_l, P_r))\!)^\mu_G = (\Omega_{lr}, Q_c) \text{ WHERE } \begin{cases} (\!(P_l)\!)^\mu_G = (\Omega_l, Q_l) \\ (\!(LeftJoin(\{\}, P_r))\!)^{\Omega_l}_G = (\Omega_{lr}, Q_{lr}) \\ Q_c = Union(LeftJoin(Q_l, P_r), Q_{lr}) \end{cases}$$

*Example 3.11.* To illustrate, let us consider the Graph $G_{ex}$ of Figure 2. An end-user wants to retrieve all human beings, and their occupations, if they have any, with the SPARQL query $Q_{opt}$:

```
SELECT * WHERE {
  ?human wdt:P31 wd:Q5          # tp1: all human beings
  OPTIONAL {                     # opt: with or without
    ?human wdt:P106 ?occupation }} # tp2: their occupations
```

Assuming that the query execution is interrupted after the first triple of $tp_1$, the evaluation $(\!(Q_{opt})\!)^{\mu_\emptyset}_G = (\!(LeftJoin(tp_1, tp_2))\!)^{\mu_\emptyset}_G$ is:

$$(\!(Q_{opt})\!)^{\mu_\emptyset}_G = \begin{array}{l}(\emptyset, Union(LeftJoin(Extend(\{?human \rightarrow \text{wd:H1}\}), \\ \qquad\qquad\qquad Extend(\{?human \rightarrow \text{wd:H1}\}, tp_2)), \\ \qquad LeftJoin(Slice(tp_1, 1), tp_2)))\end{array}$$

After simplifying, the resulting continuation query $Q^1_{opt}$ is mostly similar to the first continuation query $Q^1_{bgp}$ since the graph to explore is identical. However, $Joins$ are replaced by $LeftJoins$ in this SPARQL query:

```
SELECT * WHERE {
  { BIND ( wd:H1 AS ?human )
    OPTIONAL { ?human wdt:P106 ?occupation }
  } UNION {
    { SELECT * WHERE { ?human wdt:P31 wd:Q5 } OFFSET 1 }
    OPTIONAL { ?human wdt:P106 ?occupation }}}
```

The end-user sends back this query $Q^1_{opt}$ to the PASSAGE endpoint. After reading the first triple of the second triple pattern $tp_2$, PASSAGE

interrupts the query execution. The continuous evaluation returns:

$$(\!(Q^1_{opt})\!)^{\mu_\emptyset}_G = \begin{array}{l}(\{\!| \{?human \rightarrow \text{wd:H1}, ?occupation \rightarrow \text{wd:O1}\} |\!\}, \\ Union(Slice(Extend(\{?human \rightarrow \text{wd:H1}\}, tp_2), 1)), \\ \qquad LeftJoin(Slice(tp_1, 1), tp_2))\end{array}$$

The end-user receives one result and a continuation query $Q^2_{opt}$ where the $LeftJoin$ of the visited graph pattern became a $Join$ since it produced a result. Therefore, the rest of the computation cannot return the left mapping alone anymore. $Q^2_{opt}$ corresponds to the following SPARQL query:

```
SELECT * WHERE {
  { SELECT * WHERE {
    BIND ( wd:H1 AS ?human ) # cannot be returned alone
    ?human wdt:P106 ?occupation } OFFSET 1 # since it became a join
  } UNION {
    { SELECT * WHERE { ?human wdt:P31 wd:Q5 } OFFSET 1 }
    OPTIONAL { ?human wdt:P106 ?occupation }}
```

After having read another mapping for $tp_2$, the first part of the union is removed as it can no longer produce any result. The continuous evaluation returns:

$$(\!(Q^2_{opt})\!)^{\mu_\emptyset}_G = \begin{array}{l}(\{\!| \{?human \rightarrow \text{wd:H1}, ?occupation \rightarrow \text{wd:O2}\} |\!\}, \\ LeftJoin(Slice(tp_1, 1), tp_2)))\end{array}$$

Then, if the evaluation continues mapping by mapping:

$$(\!(Q^3_{opt})\!)^{\mu_\emptyset}_G = \begin{array}{l}(\emptyset, Union(LeftJoin(Extend(\{?human \rightarrow \text{wd:H2}\}), \\ \qquad\qquad\qquad Extend(\{?human \rightarrow \text{wd:H2}\}, tp_2)), \\ \qquad LeftJoin(Slice(tp_1, 2), tp_2)))\end{array}$$

$$(\!(Q^4_{opt})\!)^{\mu_\emptyset}_G = \begin{array}{l}(\{\!| \{?human \rightarrow \text{wd:H2}\} |\!\}, \\ LeftJoin(Extend(\{?human \rightarrow \text{wd:H3}\}, \\ \qquad\qquad Extend(\{?human \rightarrow \text{wd:H3}\}, tp_2)))\end{array}$$

$$(\!(Q^5_{opt})\!)^{\mu_\emptyset}_G = (\{\!| \{?human \rightarrow \text{wd:H3}, ?occupation \rightarrow \text{wd:O3}\} |\!\}, \{\})$$

The continuous evaluation returned complete and correct results with 4 results, where 1 result is about a human wd:H2 without occupations.

Core SPARQL also includes filters to remove some solution mappings. We only consider expressions that do not include graph patterns in their definition, such as NOT EXISTS.

*Definition 3.12 (Filter evaluation $(\!(Filter(E, P))\!)^\mu_G)$.* The evaluation of a filter with an expression $E$ over a graph pattern $P$ is:

$$(\!(Filter(E, P))\!)^\mu_G = (Filter(E, \Omega_p), Filter(E, Q_p)) \text{ WHERE } (\!(P)\!)^\mu_G = (\Omega_p, Q_p)$$

## 3.3 Properties of Continuation Queries

By implementing continuation queries, an endpoint becomes responsive and compliant with the SPARQL standard. It avoids convoy effects by interrupting query executions when they reach the timeout threshold. However, it is crucial that it provides correct and complete results when executing the sequence of continuation queries.

THEOREM 3.13 (CORRECTNESS). *Given a SPARQL query $Q = Q^0_c$, a sequence of continuation queries, $Q^1_c, \ldots, Q^n_c$ where $Q^{i+1}_c$ is the continuation query of $Q^i_c$, and their associated partial solution mappings $\Omega^i_c = \lfloor\!\lfloor Q^{i-1}_c \rfloor\!\rfloor_G$; the continuous evaluation provides complete and sound results, i.e., $[\![Q]\!]_G = \uplus_{1 \leq i \leq n} \Omega^i_c$.*

PROOF. Please refer to Appendix A.                                          □

For end-users to acknowledge completeness of results, and to bound the overhead in terms of generated traffic, two issues remain: (i) the sequence of continuation queries must be finite, (ii) and the size of generated queries must be bounded.

**Theorem 3.14 (Termination).** *Assuming a sequence of continuations queries $Q_c^1, \ldots Q_c^{i+1}$, this sequence is finite – the continuous evaluation terminates – when at least one triple pattern in each continuation continuation query $Q_c^i$ progressed by returning at least one mapping.*

**Proof.** Please refer to Appendix B. □

To bound the overhead in terms of generated traffic, the size of each continuation query must be bounded as well. However, as illustrated in Examples 3.9 and 3.11, the continuation query can comprise more triple patterns than its partially executed query. $Q_{bgp}$ has 2 triple patterns, but its first continuation query $Q_{bgp}^1$ has 3 triple patterns. We focus our analysis on a one-mapping-at-a-time evaluation strategy where the pipeline of iterators in charge of executing the query contains at most 1 mapping at any time.

**Theorem 3.15 (Continuation Query Size).** *Given a query $Q$ of $size(Q) = n$ triple patterns, the size of its continuation query $Q_c$ satisfies $size(Q_c) = O(\frac{n \cdot (n+1)}{2})$ if the bag of environment mappings $|\Omega|$ used in $Q_c$ to pass solution mappings from the left operand to the right operand of joins and optionals satisfies $|\Omega| \leq 1$.*

**Proof.** Please refer to Appendix C. □

Theorem 3.15 states that the continuation query size is quadratically upper-bounded by the number of triple patterns of the previous query. Therefore, to limit the expansion of continuation queries, passage always evaluates the smallest branch of a union in terms of the number of triple patterns. Using this smallest-branch-first evaluation strategy, every continuation query is bounded by the number of triple patterns in the *initial* query. For example, in $Q_{bgp}^1$:

$$
(\![Q_{bgp}^1]\!)_G^{\mu_\emptyset} = \begin{aligned} &(\{\![ \{?human \to \text{wd:H1}, ?occupation \to \text{wd:O1}\} ]\!\}, \\ &Union(Join(Slice(tp_1, 1), tp_2), \\ &\qquad Extend(\{?human \to \text{wd:H1}\}, Slice(tp_2, 1)))) \end{aligned}
$$

passage evaluates $Extend(\ldots, tp_2)$ in priority. If interrupted during this execution, it does not generate additional triple patterns in the continuation query. If this triple pattern is executed completely, it is removed from the continuation query. Then, passage prioritizes the second branch of the union that comprises 2 triple patterns. If the query execution is interrupted, its continuation query comprises a union between 1 triple pattern (replacing the one that was completed), and 2 triple patterns (the current ones). passage's continuations queries of $Q_{bgp}$ never comprise more than $\frac{size(Q_{bgp}) \cdot (size(Q_{bgp})+1)}{2} = 3$ triples patterns with a one-mapping-at-a-time execution.

The one-mapping-at-a-time smallest-branch-first execution of passage ensures that the size of every continuation query $Q_c$ is bounded by the size of *initial* query. The next section on experimentations empirically demonstrates that not only execution times of passage are on par with that of Blazegraph, but the size of continuation queries remain small over the continuous executions.

## 4 Related Work

Different approaches have been developed to build SPARQL engines that guarantee both responsiveness and completeness:

**SPARQL endpoints.** They follow the SPARQL protocol[1], which *describes a means for conveying SPARQL queries and updates to a SPARQL processing service and returning the results via HTTP to the entity that requested them.* Without setting quotas and by using a first-come first-served execution policy [11], SPARQL endpoints are subject to *convoy effects* [9]: one long-running query occupies the server resources and prevents other queries from executing, leading to long waiting time and degraded average completion time for queries. From the queuing theory [23] with an M/G/1 queue, the variance of the service time (job duration) directly impacts the mean waiting time in a queue. Higher variance leads to longer waiting times. Enforcing quotas in time on SPARQL endpoints reduces the variance of query execution time and consequently improves the responsiveness of the the SPARQL endpoint.

Unfortunately, if quota enforcement preserves the SPARQL endpoint responsiveness, SPARQL endpoints with quotas are no longer delivering complete results and may waste computing resources to provide partial results.

**Restricted-interface servers.** Linked Data Fragments (LDF) [14, 27] restrict a server interface to a fragment of the SPARQL algebra. Therefore, query processing must be distributed between a smart client as Comunica[26] in charge of operators not implemented by the LDF server.

With Triple Pattern Fragments (TPF) [27], the LDF server only serves pages of triple patterns. A smart client [26] decomposes the SPARQL query into a sequence of triple pattern queries evaluated by the TPF server. The TPF server ensures responsiveness and completeness as serving one page of results of a triple pattern can be done in a bounded time thanks to adequate indexation. However, compared to a traditional SPARQL engine, the execution time of queries is significantly impacted due to the intense data transfer between the smart client and the TPF server [19]. With brTPF [16], the server can process a union of triple patterns. The smart client can ask for several mappings at a time, significantly reducing the number of calls to the server. With Star Pattern Fragments (SPF) [1], the server can process star queries. The smart client can decompose SPARQL queries into star queries and process them on the SPF server. This strategy significantly reduces the data transfer between the SPF server and the smart client. Whatever the approach TPF, brTPF, or SPF, the smart client may perform joins on the client side, generating high data transfer with a substantial impact on performance compared to a traditional SPARQL engine. Moreover, LDF servers are not compliant with the SPARQL endpoint standard, limiting the adoption. Compared to LDF, passage operates as a SPARQL endpoint and supports continuations for core SPARQL operators. By declaring passage's interface in Comunica [26], end-users can execute any SPARQL query, i.e., Comunica decomposes the query such that only core SPARQL subqueries are sent to passage.

Smart-KG [7] and WiseKG [6] combine an LDF server with predefined and compressed partitions of triples. Thanks to a cost model,

---

[1]https://www.w3.org/TR/2013/REC-sparql11-protocol-20130321/

**Table 1: Average total execution time (min) for BGPs and OPTs queries with different CPU configurations. Total represents the sum of execution times of both BGPs and OPTs.**

| | 1vCPU | | | 4vCPU | | |
|---|---|---|---|---|---|---|
| | **Total** | **BGPs** | **OPTs** | **Total** | **BGPs** | **OPTs** |
| **Blazegraph** | 238.86 | 141.17 | 97.69 | 89.21 | 51.49 | 37.72 |
| **Apache Jena** | 610.06 | 347.76 | 262.30 | 611.41 | 347.41 | 264.00 |
| **SaGe** | 407.12 | 277.92 | 129.20 | 409.11 | 278.83 | 130.28 |
| PASSAGE | 143.72 | 86.58 | 57.14 | 130.52 | 79.96 | 50.56 |
| SaGe 60s | 413.81 | 279.02 | 134.79 | 413.17 | 278.65 | 134.52 |
| PASSAGE 60s | 145.30 | 87.38 | 57.92 | 132.24 | 80.94 | 51.30 |

**Table 2: Average size of continuation queries or saved plans, and average number of continuation queries (#TO) for PASSAGE 60s or average number of suspended plans for SaGe 60s.**

| | | BGPs | | OPTs | |
|---|---|---|---|---|---|
| | | size | #TO/Total | size | #TO/Total |
| 1vCPU | SaGe 60s | 1.182kB | 5.177/**292.67** | 1.927kB | 3.070/**152.67** |
| | PASSAGE 60s | 0.821kB | 1.492/**62.67** | 0.827kB | 1.556/**41.67** |
| 4vCPU | SaGe 60s | 1.168kB | 5.197/**296** | 1.929kB | 3.044/**153.67** |
| | PASSAGE 60s | 0.780kB | 1.390/**57** | 0.649kB | 1.372/**30.33** |

a smart client can process queries between TPF or SPF and download a partition of triples from the server for processing on a smart client. Using pre-defined and compressed partitions of triples can significantly reduce the data transfer between an LDF server and a smart client.

While using pre-defined partitions is an effective optimization technique that can be used by any smart client, it is difficult to integrate into a standard SPARQL.

**Web preemption.** SaGe suspends SPARQL queries after a quantum of time to return partial results with a saved physical plan [2, 12, 19]. The physical plan can be reloaded by the server, allowing it to restart from where it had been stopped. SaGe ensures responsiveness and completeness for BGPs, some aggregate queries[12], and a subset of property path queries[2]. However, SaGe is not compliant with the SPARQL endpoint standard. It requires to extend the interface of the server for re-loading saved plans.

In PASSAGE, saved plan are replaced by continuation queries that are just regular SPARQL queries. When a saved plan is encoded and compressed, a continuation query is just human readable. When a new server interface is required to reload saved plan, a continuation query is just executed as new SPARQL query, maybe with a plan different from its previous query. Compared to SaGe, continuation queries are defined in a formal framework allowing to express the continuation problem, proving its correctness, its termination and computing space complexities. Web preemption is defined at the physical level, continuation queries are defined at the logical level.

## 5 Experiments Study

This experimental study aims to answer three questions empirically:

(1) What is the difference in execution time between PASSAGE with quotas and PASSAGE without quotas?
(2) What are the average size and number of generated continuation queries?
(3) How does the execution time of queries with PASSAGE compare to that of representative SPARQL engines?

PASSAGE is implemented in Java on top of Blazegraph's storage and supports core SPARQL operators. To ensure logarithmic access times to triple patterns by offset, we rely on the augmented balanced trees index of Blazegraph[2]. Additionally, we extended the smart client Comunica [26] to decompose queries for PASSAGE, i.e., SPARQL operators not supported by PASSAGE are executed within Comunica. The code for reproducible experiments is publicly available on GitHub at: https://anonymous.4open.science/r/passage-experiments-C0D3.

### 5.1 Experimental Setup

**Datasets:** We use the WDBench [3], a real-world large dataset extracted from Wikidata containing around 1.25 billion triples.

**Queries:** We focus on basic graph pattern queries (BGPs) and optional queries (OPTs) from WDBench that fail to complete within a 60-second timeout on Blazegraph, as those terminating within 60 seconds are considered to be adequately handled by current SPARQL engines. Specifically, we randomly selected 49 BGPs and 38 OPTs, which take between one to five minutes to execute on Blazegraph using a single vCPU core with 54 GB of RAM. The selected BGPs contain 2 to 6 triple patterns, while the OPTs contain 2 to 19 triple patterns.

**Approaches:** We compare the following approaches:
- PASSAGE: We run PASSAGE query engine the Blazegraph storage without a timeout and with a 60-second timeout, denoted PASSAGE and PASSAGE-60s, respectively. The 60 seconds was chosen to comply with Wikidata's fair-use policy.
- Blazegraph: We run Blazegraph (v 2.1.4), a high-performance SPARQL engine currently serving Wikidata, without any quota or limitations.
- Apache Jena: A popular open-source framework for building Semantic Web and Linked Data applications[3]. Jena (v 5.1.0) runs with its TDB2 data storage, and we set its query execution timeout to 10 minutes.
- SaGe [19]: A SPARQL query engine based on Web Preemption. We run the SaGe query engine without a time quantum and with a 60-second time quantum, denoted SaGe and SaGe-60s, respectively. The data is stored in read-only HDT files [10].

We did not include TPF [27] or brTPF [13] as they are already outperformed by SaGe [19]. Similarly, we excluded Smart-KG [7] and WiseKG [6] because they require pre-computing partitions of triples for query processing. Serving partitions of triples is not part of the SPARQL endpoint protocol and is irrelevant in our context.

**Hardware Configuration:** We run all the servers on a local cloud instance with Ubuntu 20.04.4.LTS, an AMD EPYC 7513-Core processor with 16 vCPUs allocated to the VM, 1 TB SSD, and 64 GB of RAM. To ensure a fair resource distribution among the different approaches, we conducted experiments using two different

---

[2]https://github.com/blazegraph/database/wiki/BTreeGuide
[3]https://jena.apache.org/

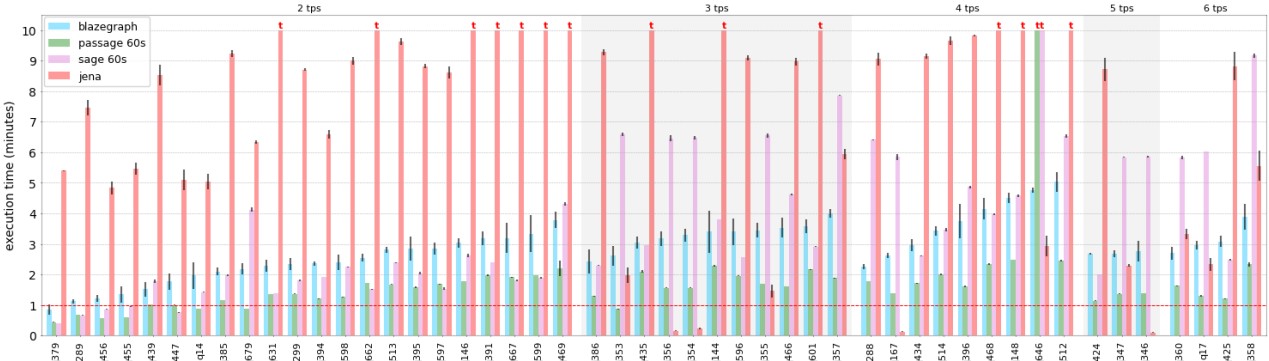

**Figure 3: Execution time for BGPs queries group by number of triple patterns in the query with 1vCPU.**

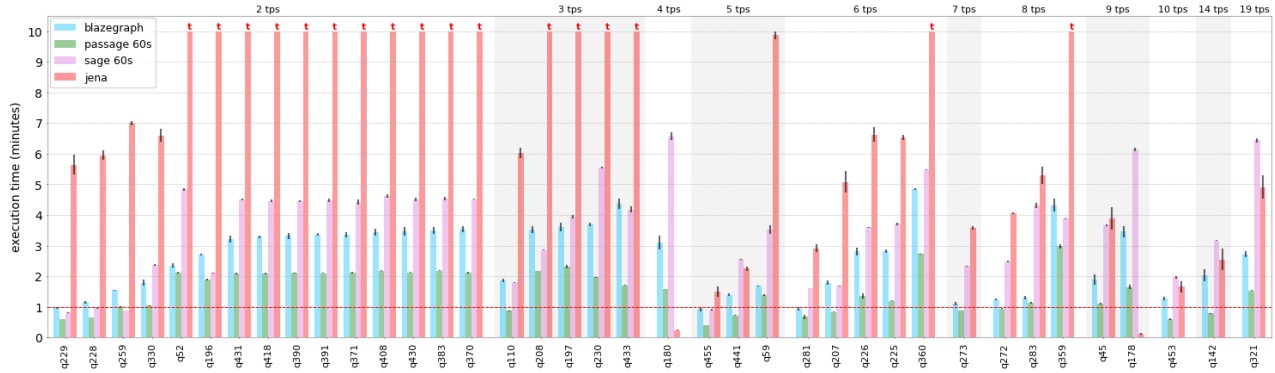

**Figure 4: Execution time for OPTs queries group by number of triple patterns in the query with 1vCPU.**

configurations: one with a single virtual CPU (1vCPU) and another with four virtual CPUs (4vCPU), both with 54 GB of RAM allocated to the virtual machine. The 1vCPU setup servers as a fair baseline for all engines, particularly those that do not utilize parallelism, ensuring no engine gains an unfair advantage from multi-threading. The 4vCPU setup allows us to evaluate the potential improvement from increased computational resources and parallel processing.

**Evaluation Metrics:** We always ensure that our approach pas-sage produces complete results using Blazegraph's results as ground truth. Presented results correspond to the average obtained of three successive executions of the queries workloads. We measure:

- Total workload execution time: is the total time the engine takes to execute queries workload and get complete results.
- Number of continuation queries: is the number of continuations for each query.
- Size of continuation queries: is the size of a continuation query in kilobytes.

## 5.2 Experimental Results

*What is the difference in execution time between passage with quotas and passage without quotas?*

Table 1 presents the average execution time for different query workloads across the two vCPU configurations, comparing various

approaches. The difference in execution time between passage-60s and passage for 1vCPU is 1.58 minutes. The total number of generated continuation queries is approximately 104 (as shown in Table 2). By dividing the 1.58 minutes overhead by 104, we find the overhead per continuation query to be nearly 900ms.

Applying the same analysis for SaGe and SaGe-60s, the overhead for interrupting queries in the 1vCPU setup amounts to 6.69 minutes over 413 minutes of execution. The total number of interruptions for SaGe-60s for this workload is approximately 589, resulting in an overhead of about 681 milliseconds per interruption in SaGe, which is slightly less than that of passage-60s. Compared to SaGe, the continuation queries in passage-60s require parsing and optimization, which explains the slight difference in time per interruption. However, re-optimizing continuation queries may lead to better execution plans, representing a good trade-off in performance.

*What are the average size and number of generated continuation queries?*

Table 2 presents the average size and number of continuation queries for BGPs and OPTs queries across different configurations. For passage-60s, regardless of the configuration, the average number of continuation queries remains low, always fewer than 2. In

contrast, SaGe-60s generates more saved plans; for instance, it generates an average of 5 for BGPs queries. As PASSAGE-60s is faster than SaGe-60s, PASSAGE-60s is less interrupted than SaGe-60s.

Regarding query size, PASSAGE-60s consistently produces small query sizes, averaging around 0.8 KB, irrespective of the workload or configuration. In comparison with SaGe-60s, the size of the suspended plans are larger than the size of continuation queries of PASSAGE-60s, whatever the setup.

*How does the execution time of queries with PASSAGE compare to that of representative SPARQL engines?*

Table 1 presents the performance of PASSAGE-60s with SaGe-60s, Blazegraph, and Apache Jena. The quota of 60s applies only to PASSAGE-60s and SaGe-60s, which ensure both completeness and responsiveness. BlazeGraph and Apache Jena execute the workload without interruption, i.e., without quota. While they ensure the completeness, they do not guarantee the responsiveness.

On the 1vCPU configuration, PASSAGE-60s is the fastest engine for BGPs and OPTs queries. It is slightly faster than Blazegraph but approximately 3 times faster than Sage-60s and 4 times faster than Jena. On the 4vCPU configuration, Blazegraph is the best-performing engine. Blazegraph implements intra-query parallelism and takes advantage of the 4vCPU while other engines continue to use mainly 1vCPU. JENA and SaGe-60s have very similar execution time with 1vCPU, PASSAGE-60s execution time is slightly improved.

For further detail, Figures 3, and 4 illustrate execution time per query for 1vCPU for BGPs and OPTs queries, respectively. In these figures, queries are grouped by the number of triple patterns labeled along the x-axis, while the y-axis shows the average execution time through 3 runs in minutes, ranging from zero to ten. The queries are ordered by Blazegraph's execution time within each group. Red "t" marks appear above some bars, indicating those queries timed out on the engine. The horizontal red line at 1min represents the quota of 60s.

Figure 3 presents the result for BGPs queries. Blazegraph executes a query within 1min and 5min. Jena's execution times are significantly higher for many queries than other engines, with frequent timeouts at 10 minutes (12 out of 49 queries), likely due to join order. SaGe-60s can terminate all the workload queries except query q646 due to poor join ordering.

In contrast, our engine, PASSAGE-60s, delivers good overall performance with an average execution time of 106.02 seconds and only one timeout on a complex query (q646). This is also caused by the join order; after fixing the join order, the execution time dropped drastically from over 10 min to just 3 seconds. While Jena outperforms PASSAGE in a few isolated cases, these are exceptions due to specific join orders.

For OPTs queries in Figure 4, Jena times out after 10 min on nearly half of the workload (17 queries out of 38), though it occasionally performs better than PASSAGE-60s on 2 queries due to efficient join order. PASSAGE consistently outperforms all other engines, with an average execution time of 90.21 seconds, completing 13 queries in less than 60 seconds.

The detailed results for 4vCPU are available in Appendix D.

Overall, we observe that PASSAGE consistently delivers high performance in the same order of magnitude as Blazegraph while ensuring results completeness and responsiveness.

# 6 Conclusion and Future Work

In this paper, we introduced the concept of SPARQL continuation query to enable public SPARQL endpoints to achieve completeness, responsiveness and performance without wasting computation resources. When query execution reaches the timeout, the endpoint interrupts it, returns partial results along with a continuation query that efficiently represents the missing results in a SPARQL-compliant format. To support continuations, PASSAGE relies on two key assumptions: (1) Triple pattern evaluations must return an ordered list of mappings, a condition easily met when RDF triples are indexed by traditional B-trees. (2) Each partial evaluation of a continuation query must ensure progress, meaning it processes at least one scan of any triple pattern of the query. This requires efficient access to an offset in the triple patterns, a feature already supported in Blazegraph storage and HDT. With these two requirements in place, any SPARQL engine can be transformed into a continuous SPARQL engine capable of providing completeness, responsiveness, and robust performance, as demonstrated in our experiments.

For future work, we plan to work on intra-query parallelism for continuations queries to speed up the execution of core SPARQL queries. Additionally, we plan to support more SPARQL clauses such as COUNT, GROUP BY, and DISTINCT, gradually deporting smart clients operations to servers to enhance performance.

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

## A  Proof of Correctness

Proof. Given a solution to the continuation queries problem, $Q_c^1, \ldots, Q_c^n$, and its associated solution mappings $\Omega_c^i = \llbracket Q_c^{i-1} \rrbracket_G$

(with $Q_c^0 = Q$), it must hold that $\llbracket Q \rrbracket_G = \biguplus_{1 \le i \le n} \Omega_c^i$. We must demonstrate that every continuation step satisfies $\llbracket Q_c^i \rrbracket_G = \Omega_c^{i+1} \uplus \llbracket Q_c^{i+1} \rrbracket_G$. We prove it by induction on the structure of the query $Q$.

The base case corresponds to a triple pattern, $Q = tp$. By Definition 3.3, its evaluation is:

$$\llbracket tp \rrbracket_G^\mu = \{\!\{ \mu_1, \ldots \mu_i \}\!\} \uplus \llbracket Slice(Extend(\mu, tp), i) \rrbracket_G$$
$$\Leftrightarrow \qquad \{\text{Definitions 3.3 and 3.4}\}$$
$$\{\!\{ \mu_1, \ldots \mu_k \}\!\} = \{\!\{ \mu_1, \ldots \mu_i \}\!\} \uplus \{\!\{ \mu_{i+1}, \ldots \mu_k \}\!\}$$

For the inductive case $Q = Join(P_l, P_r)$, the inductive hypothesis is: $\llbracket P_l \rrbracket_G^\mu = \Omega_l \uplus \llbracket P_{l_c} \rrbracket_G^{\Omega_l}$ and $\llbracket P_r \rrbracket_G^{\Omega_l} = \Omega_{lr} \uplus \llbracket P_{r_c} \rrbracket_G^{\Omega_l}$. Therefore:

$\llbracket Join(P_l, P_r) \rrbracket_G^\mu = \{\text{Definition 3.7}\}$

$\qquad \Omega_{lr} \uplus \llbracket Union(Join(P_{l_c}, P_r), Extend(\Omega_l, P_{r_c})) \rrbracket_G$

$= \{\text{definitions of } Union, Join, \text{ and } Extend\}$

$\qquad \Omega_{lr} \uplus (\llbracket Pl_c \rrbracket_G \bowtie \llbracket Pr \rrbracket_G) \uplus \llbracket Pr_c \rrbracket_G^{\Omega_l}$

$= \{\text{inductive hypothesis}\}$

$\qquad \llbracket P_r \rrbracket_G^{\Omega_l} \uplus (\llbracket P_{l_c} \rrbracket_G \bowtie \llbracket P_r \rrbracket_G)$

$= \{\text{by } \llbracket P_r \rrbracket_G^{\Omega_l}\}$

$\qquad (\Omega_l \bowtie \llbracket P_r \rrbracket_G) \uplus (\llbracket P_{l_c} \rrbracket_G \bowtie \llbracket P_r \rrbracket_G)$

$= \{\text{distributivity join over union}\}$

$\qquad (\Omega_l \uplus \llbracket P_{l_c} \rrbracket_G) \bowtie \llbracket P_r \rrbracket_G)$

$= \{\text{inductive hypothesis}\}$

$\qquad \llbracket P_l \rrbracket_G^\mu \bowtie \llbracket P_r \rrbracket_G$

For the inductive case $Q = LeftJoin(\{\}, P_r)$, the inductive hypothesis is: $\llbracket P_r \rrbracket_G^{\Omega_l} = \Omega_r \uplus \llbracket P_{r_c} \rrbracket_G^{\Omega_l}$ and we should demonstrate that: $\llbracket LeftJoin(\{\}, P_r) \rrbracket_G = \llparenthesis LeftJoin(\{\}, P_r) \rrparenthesis_G \uplus \llbracket LeftJoin(\{\}, P_r)_c \rrbracket_G$. By Definition 3.10, there are four cases for $\llparenthesis LeftJoin(\{\}, P_r) \rrparenthesis_G$ and $LeftJoin(\{\}, P_r)_c$:

(Case 1) $\llparenthesis LeftJoin(\{\}, P_r) \rrparenthesis_G = \{\!\{ \mu \}\!\}$ and $LeftJoin(\{\}, P_r)_c = \{\}$ with $\llbracket P_r \rrbracket_G^\mu = \emptyset \uplus \llbracket \{\} \rrbracket_G$:

$\llbracket LeftJoin(\{\}, P_r) \rrbracket_G = \{\text{Definition 3.10}\}$

$\qquad \llparenthesis LeftJoin(\{\}, P_r) \rrparenthesis_G \uplus \llbracket LeftJoin(\{\}, Pr)_c \rrbracket_G$

$= \{\text{Case 1}\}$

$\qquad \{\!\{ \mu \}\!\} \uplus \llbracket \{\} \rrbracket_G$

$= \{\text{inductive hypothesis}\}$

$\qquad \{\!\{ \mu \}\!\} \bowtie \llbracket P_r \rrbracket_G^\mu$

(Case 2) $\llparenthesis LeftJoin(\{\}, P_r) \rrparenthesis_G = \{\!\{ \mu \}\!\} \bowtie \Omega_r$ and $LeftJoin(\{\}, P_r)_c = \{\}$ given that $\llbracket P_r \rrbracket_G^\mu = \Omega_r \uplus \llbracket \{\} \rrbracket_G$:

$\llbracket LeftJoin(\{\}, P_r) \rrbracket_G = \{\text{Definition 3.10}\}$

$\qquad \llparenthesis LeftJoin(\{\}, P_r) \rrparenthesis_G \uplus \llbracket LeftJoin(\{\}, P_r)_c \rrbracket_G$

$= \{\text{Case 2}\}$

$\qquad \{\!\{ \mu \}\!\} \bowtie \Omega_r \uplus \llbracket \{\} \rrbracket_G$

$= \{\text{definition of } \{\}\}$

$\qquad \{\!\{ \mu \}\!\} \bowtie \Omega_r$

$= \{\text{Case 2 and inductive hypothesis}\}$

$\qquad \{\!\{ \mu \}\!\} \bowtie \llbracket P_r \rrbracket_G$

(Case 3) $\lfloor LeftJoin(\{\}, P_r)\rfloor_G = \emptyset$ and $LeftJoin(\{\}, P_r)_c = LeftJoin(Extend(\mu), P_{r_c})$ given that $[\![P_r]\!]_G^\mu = \emptyset \uplus [\![P_{r_c}]\!]_G$:

$$[\![LeftJoin(\{\}, P_r)]\!]_G = \{\text{Definition 3.10}\}$$
$$\lfloor LeftJoin(\{\}, P_r)\rfloor_G \uplus [\![LeftJoin(\{\}, P_r)_c]\!]_G$$
$$= \{\text{Case 3}\}$$
$$\emptyset \uplus [\![LeftJoin(Extend(\mu), P_{r_c})]\!]_G$$
$$= \{\text{definitions of } LeftJoin \text{ and } Extend\}$$
$$\{\!\{\mu\}\!\} \bowtie [\![P_{r_c}]\!]_G$$
$$= \{\text{Case 3 and inductive hypothesis}\}$$
$$\{\!\{\mu\}\!\} \bowtie [\![P_r]\!]_G$$

(Case 4) $\lfloor LeftJoin(\{\}, P_r)\rfloor_G = \Omega_r$ and $LeftJoin(\{\}, P_r)_c = P_{r_c}$ with $[\![P_r]\!]_G^\mu = \Omega_r \uplus [\![P_{r_c}]\!]_G$ and $|\Omega_r| > 0$:

$$[\![LeftJoin(\{\}, Pr)]\!]_G = \{\text{Definition 3.10}\}$$
$$\lfloor LeftJoin(\{\}, P_r)\rfloor_G \uplus [\![LeftJoin(\{\}, P_r)_c]\!]_G$$
$$= \{\text{Case 4}\}$$
$$\Omega_r \uplus [\![P_{r_c}]\!]_G$$
$$= \{\text{Case 4 and inductive hypothesis}\}$$
$$[\![P_r]\!]_G^\mu$$
$$= \{\text{by } [\![P_r]\!]_G^\mu\}$$
$$\{\!\{\mu\}\!\} \bowtie [\![P_r]\!]_G$$
$$= \{\text{inductive hypothesis}\}$$
$$\{\!\{\mu\}\!\} \bowtie (\Omega_r \uplus [\![P_{r_c}]\!]_G)$$
$$= \{\text{by } |\Omega_r| > 0 \text{ and } \Omega_r = \lfloor P_r\rfloor_G^\mu\}$$
$$(\{\!\{\mu\}\!\} \bowtie (\Omega_r \uplus [\![P_{r_c}]\!]_G)) \uplus (\{\!\{\mu\}\!\} \backslash (\Omega_r \uplus [\![P_{r_c}]\!]_G))$$
$$= \{\text{definition of } LeftJoin\}$$
$$\{\!\{\mu\}\!\} \bowtie (\Omega_r \uplus [\![P_{r_c}]\!]_G)$$
$$= \{\text{inductive hypothesis}\}$$
$$\{\!\{\mu\}\!\} \bowtie [\![P_r]\!]_G$$

Given that we have proved for each of the four cases that $\lfloor LeftJoin(\{\}, P_r)\rfloor_G \uplus [\![LeftJoin(\{\}, P_r)_c]\!]_G = \{\!\{\mu\}\!\} \bowtie [\![P_r]\!]_G$, we have shown that $Q_c^i = LeftJoin(\{\}, P_r)$ satisfies $[\![Q_c^i]\!]_G = \Omega_c^{i+1} \uplus [\![Q_c^{i+1}]\!]_G$

For the inductive case $Q = Q_o = LeftJoin(P_l, P_r)$, the inductive hypothesis is: $[\![P_l]\!]_G^\mu = \Omega_l \uplus [\![P_{l_c}]\!]_G$ and $[\![LeftJoin(\{\}, P_r)]\!]_G^{\Omega_l} = \Omega_{lr} \uplus [\![LeftJoin(\{\}, P_r)_c]\!]_G^{\Omega_l}$. Therefore:

$$[\![Q_o]\!]_G^\mu$$
$$= \{\text{Definition 3.10}\}$$
$$\Omega_{lr} \uplus [\![Union(LeftJoin(P_{l_c}, P_r), Extend(\Omega_l, LeftJoin(\{\}, P_r)_c))]\!]_G$$
$$= \{\text{definitions of } Union, LeftJoin, \text{ and } Extend\}$$
$$\Omega_{lr} \uplus [\![P_{l_c}]\!]_G \bowtie [\![P_r]\!]_G \uplus \Omega_l \bowtie [\![LeftJoin(\{\}, P_r)_c]\!]_G$$

The values of $\Omega_{lr}$ and $LeftJoin(\{\}, P_r)_c$ follow the four cases detailed for $[\![LeftJoin(\{\}, P_r)]\!]_G$, so we shall split $\Omega_l$ into four bags $\Omega_1 \ldots \Omega_4$ where each of these bags includes all the $\mu$s that fall into each of those four cases:

For $\Omega_1$: $[\![LeftJoin(\{\}, P_r)]\!]_G^{\Omega_1} = \Omega_1 \uplus [\![\{\}]\!]_G$;
For $\Omega_2$: $[\![LeftJoin(\{\}, P_r)]\!]_G^{\Omega_2} = \Omega_2 \bowtie \Omega_r \uplus [\![\{\}]\!]_G$;

For $\Omega_3$: $[\![LeftJoin(\{\}, P_r)]\!]_G^{\Omega_3} = \emptyset \uplus [\![LeftJoin(Extend(\Omega_3), P_{r_c})]\!]_G$;
For $\Omega_4$: $[\![LeftJoin(\{\}, P_r)]\!]_G^{\Omega_4} = \Omega_r \uplus [\![P_{r_c}]\!]_G$.
Moreover: $\Omega_{lr} = \Omega_1 \uplus (\Omega_2 \bowtie \Omega_r) \uplus \Omega_r$; and
$[\![LeftJoin(\{\}, P_r)_c]\!]_G = [\![LeftJoin(Extend(\Omega_3), P_{r_c})]\!]_G \uplus [\![P_{r_c}]\!]_G$.

$$[\![Q_o]\!]_G^\mu$$
$$= \{\text{inductive hypotheses and split described above}\}$$
$$[\![LeftJoin(\{\}, P_r)]\!]_G^{\Omega_1} \uplus [\![LeftJoin(\{\}, P_r)]\!]_G^{\Omega_2} \uplus [\![LeftJoin(\{\}, P_r)]\!]_G^{\Omega_3}$$
$$\uplus [\![LeftJoin(\{\}, P_r)]\!]_G^{\Omega_4} \uplus [\![P_{l_c}]\!]_G \bowtie [\![P_r]\!]_G$$
$$= \{\text{split}\}$$
$$[\![LeftJoin(\{\}, P_r)]\!]_G^{\Omega_l} \uplus [\![P_{l_c}]\!]_G \bowtie [\![P_r]\!]_G$$
$$= \{\text{definition of } LeftJoin\}$$
$$\Omega_l \bowtie [\![\{\}]\!]_G \bowtie [\![P_r]\!]_G \uplus [\![P_{l_c}]\!]_G \bowtie [\![P_r]\!]_G$$
$$= \{\text{evaluation of } \{\} \text{ and distributivity}\}$$
$$(\Omega_l \uplus [\![P_{l_c}]\!]_G) \bowtie [\![P_r]\!]_G$$
$$= \{\text{inductive hypotheses}\}$$
$$[\![P_l]\!]_G \bowtie [\![P_r]\!]_G$$

Given that we have proved
$\lfloor LeftJoin(P_l, P_r)\rfloor_G \uplus [\![LeftJoin(P_l, P_r)_c]\!]_G = [\![P_l]\!]_G \bowtie [\![P_r]\!]_G$, we have shown that $Q_c^i = LeftJoin(P_l, P_r)$ satisfies $[\![Q_c^i]\!]_G = \Omega_c^{i+1} \uplus [\![Q_c^{i+1}]\!]_G$

The inductive cases $Q = Union(P_l, P_r)$, $Q = Extend(\mu, P)$, and $Q = Filter(E, P)$ are similar as they depend only on the continuation query of their operands and therefore, we detail only the first one.

For the inductive case $Q = Q_u = Union(P_l, P_r)$, the inductive hypothesis is: $[\![P_l]\!]_G^\mu = \Omega_l \uplus [\![P_{l_c}]\!]_G$ and $[\![P_r]\!]_G^\mu = \Omega_r \uplus [\![P_{r_c}]\!]_G$. Therefore:

$$[\![Q_u]\!]_G^\mu$$
$$= \{\text{Definition 3.8}\}$$
$$\Omega_l \uplus \Omega_r \uplus [\![Union(P_{l_c}, P_{r_c})]\!]_G$$
$$= \{\text{definitions of } Union\}$$
$$\Omega_l \uplus \Omega_r \uplus [\![P_{l_c}]\!]_G \uplus [\![P_{r_c}]\!]_G$$
$$= \{\text{hypothesis inductive}\}$$
$$[\![P_l]\!]_G^\mu \uplus [\![P_r]\!]_G^\mu$$

Therefore $Q_c^i = Union(P_l, P_r)$ satisfies $[\![Q_c^i]\!]_G = \Omega_c^{i+1} \uplus [\![Q_c^{i+1}]\!]_G$

As we have shown that $[\![Q_c^i]\!]_G = \Omega_c^{i+1} \uplus [\![Q_c^{i+1}]\!]_G$ is satisfied in the base case and the inductive cases, we have completed the proof of Theorem 3.13.

□

# B Proof of Termination

PROOF. Given $Q = Q_c^0$ and $Q_c^{i+1}$ the continuation query of $Q_c^i$, the sequence of continuation queries $Q_c^1, \ldots Q_c^n$ is finite if the space explored by $Q_c^{i+1}$ is strictly smaller than the space explored by $Q_c^i$. Given the definition of continuation queries, this condition does not hold in general, but only for executions were each partial execution is able to make some progress in the evaluation of the query.
We must prove that the space explored does not increase over continuations, i.e., $space(Q_c^{i+1}) \leq space(Q_c^i)$, then identify the restrictions needed to ensure that an execution makes progress.

First, we define the space explored by a graph pattern as follows:

$$space(tp) = card(\llbracket tp \rrbracket_G)$$
$$space(Slice(Extend(\mu, tp), i)) = max(card(\llbracket tp \rrbracket_G \bowtie \mu) - i, 0)$$
$$space(Join(P_l, P_r)) = space(P_l) \cdot space(P_r)$$
$$space(LeftJoin(P_l, P_r)) = space(Join(P_l, P_r))$$
$$space(Union(P_l, P_r)) = space(P_l) + space(P_r)$$
$$space(Filter(E, P)) = space(P)$$
$$space(Extend(\Omega, P)) = (1/(2 + |\Omega|)) \cdot space(P)$$

For the base case, $Q_i$ is a triple pattern $tp$, its continuation query is $Slice(Extend(\mu, tp), i)$. This continuation query explores a smaller space when $i$ is greater than zero and the same space when $i$ is zero.

The inductive cases $Join(P_l, P_r)$ and $LeftJoin(P_l, P_r)$ are similar and we detail only the first one. We have as inductive hypothesis that $space(P_{l_c}) \leq space(P_l)$ and $space(P_{r_c}) \leq space(P_r)$. For a conjunctive query $Q_j = Join(P_l, P_r)$, we have the continuation query $Q_{j_c} = Union(Join(P_{l_c}, P_r), Extend(\Omega_l, P_{r_c}))$. By definitions of the respective space for $Union$, $Join$, and $Extend$:

$$space(Q_{j_c}) = space(P_{l_c}) \cdot space(P_r) + (1/(2 + |\Omega_l|)) \cdot space(P_{r_c})$$

For $space(Q_{j_c})$ to be smaller than $space(P_l) \cdot space(P_r)$, we identify a restriction: $space(P_{l_c})$ must be at most $space(P_l) - 1$. Therefore:

$$space(P_{l_c}) \cdot space(P_r) + (1/(2 + |\Omega_l|)) \cdot space(P_{r_c})$$
$$\leq \quad \{constraint\}$$
$$(space(P_l) - 1) \cdot space(Pr) + (1/(2 + |\Omega_l|)) \cdot space(P_{r_c})$$
$$= \quad \{arithmetic\}$$
$$space(P_l) \cdot space(P_r) - space(P_r) + (1/(2 + |\Omega_l|)) \cdot space(P_{r_c})$$

For this space to be smaller than $space(P_l) \cdot space(P_r)$, it is sufficient that $(1/(2 + |\Omega_l|)) \cdot space(P_{r_c}) < space(P_r)$ and this follows from the inductive hypothesis.

The inductive cases $Union(P_l, P_r)$, $Filter(E, P)$, and $Extend(\mu, P)$ are similar and we detail only the first one. The continuation query of $Union(P_l, P_r)$ is $Union(P_{l_c}, P_{r_c})$.

$$space(Union(P_{l_c}, P_{r_c})) = space(P_{l_c}) + space(P_{r_c})$$

For this space to be smaller than the space explored by $Union(P_l, P_r)$ at least one of $space(P_{l_c}) < space(P_l)$ or $space(P_{r_c}) < space(P_r)$ must hold.

The overall restriction we need to impose to obtain a finite sequence of continuation queries is that the partial evaluation of at least one triple pattern in $Q_c^i$ has progressed by returning at least one mapping. If $Q_c^i$ is a join or optional, for these operands their left operand must have progressed. This last restriction is due to the way the join and optional are formalized, where the solution mappings from the left operand are injected into the right operand. □

## C  Proof of Query Size

Theorem 3.15 states that there is a bound on the number of triple patterns in a continuation query, i.e., the size of $Q_c$, the continuation query of $Q$ satisfies $size(Q_c) = O(\frac{size(Q) \cdot (size(Q)+1)}{2})$ given that $\Omega$ the bag of environment mappings used in $Q_c$ to pass solution mappings from the left operand to the right operands is at most 1.

PROOF. We must prove that, $size(Q_c) \leq c \cdot \frac{size(Q) \cdot (size(Q)+1)}{2}$ for some constant $c$.

First, we define the size of a graph pattern as follows:

$$size(tp) = 1$$
$$size(Slice(Extend(\mu, tp), i)) = \begin{cases} 1 \text{ IF } i < |\llbracket tp \rrbracket_G \bowtie \mu|) \\ 0 \text{ OTHERWISE} \end{cases}$$
$$size(Join(P_l, P_r)) = size(P_l) + size(P_r)$$
$$size(LeftJoin(P_l, P_r)) = size(Join(P_l, P_r))$$
$$size(Union(P_l, P_r)) = size(Join(P_l, P_r))$$
$$size(Filter(E, P)) = size(P)$$

For the base case, $Q = tp$: $size(Slice(Extend(\mu, tp), i)) \leq c$. This expression holds for $c \geq 1$ because $tp$'s continuation query is a sliced triple pattern $Slice(Extend(\mu, tp), i)$ comprising at most one triple pattern.

For the inductive cases $Q = Join(P_l, P_r)$ or $Q = LeftJoin(P_l, P_r)$ with partial results from $P_l$, $|\Omega_l|$, the inductive hypothesis is: $size(P_{l_c}) \leq c \cdot \frac{size(P_l) \cdot (size(P_l)+1)}{2}$ and $size(P_{r_c}) \leq c \cdot \frac{size(P_r) \cdot (size(P_r)+1)}{2}$. Let $l_c, r_c, l, r$ be $size(P_{l_c})$, $size(P_{r_c})$, $size(P_l)$, $size(P_r)$. According to Definitions 3.7 and 3.10 the size of the continuation query is:

$$l_c + r + r_c \cdot |\Omega_l| \leq \{inductive \ hypothesis\}$$
$$c \cdot \frac{l \cdot (l+1)}{2} + r + c \cdot \frac{r \cdot (r+1)}{2} \cdot |\Omega_l|$$
$$\leq \{arithmetic \ and \ |\Omega_l| \leq 1\}$$
$$\frac{c \cdot l^2}{2} + \frac{c \cdot l}{2} + r + \frac{c \cdot r^2}{2} + \frac{c \cdot r}{2}$$

This size must be at most $\frac{c \cdot l^2}{2} + c \cdot l \cdot r + \frac{c \cdot l}{2} + \frac{c \cdot r}{2} + \frac{c \cdot r^2}{2}$. Since $P_l$ has at least one triple pattern ($l \geq 1$), $c \geq 1$ is enough to make this last condition hold.

Given that $c \geq 1$, we have that the size of a continuation query of a $Join(P_l, P_r)$ or $LeftJoin(P_l, P_r)$ (with $size(P_l) = l$ and $size(P_r) = r$) has a size that is bound by $\frac{(l+r) \cdot (l+r+1)}{2}$.

For the inductive cases $Q = Union(P_l, P_r)$, the size of continuation query depends only on the size of the continuation queries of the operators, i.e., $size(P_{l_c}) + size(P_{r_c})$. By inductive hypothesis, this size is bound by $c \cdot \frac{size(P_l) \cdot (size(P_l)+1)}{2} + c \cdot \frac{size(P_r) \cdot (size(P_r)+1)}{2}$ and this last expression is less than $c \cdot \frac{size(P_l)^2}{2} + c \cdot \frac{size(P_l) \cdot size(P_r)}{2} + c \cdot \frac{size(P_l)}{2} + c \cdot \frac{size(P_r)}{2} + c \cdot \frac{size(P_r)^2}{2}$ if $c \cdot \frac{size(P_l) \cdot size(P_r)}{2} \geq 0$. Given that $c \geq 1$, this last expression holds and the size of the continuation query of $Union(P_l, P_r)$ (with $size(P_l) = l$ and $size(P_r) = r$) is at most $c \cdot \frac{(l+r) \cdot (l+r+1)}{2}$.

For the filter and extend, the proof is similar to the one for unions as the operand already satisfies the property. □

## D  Additional Experimental Results

Figures 5 and 6 detail the performance in execution time of all approaches in the setup with 4vCPU. As expected, Blazegraph's performance is significantly better than the 1vCPU configuration, with an average execution time of 63.05 seconds for BGPs queries and 60.09 seconds for OPTs queries. It clearly benefits from the additional computational resources and parallel processing capabilities. It surpasses all other engines across all queries, with a significant

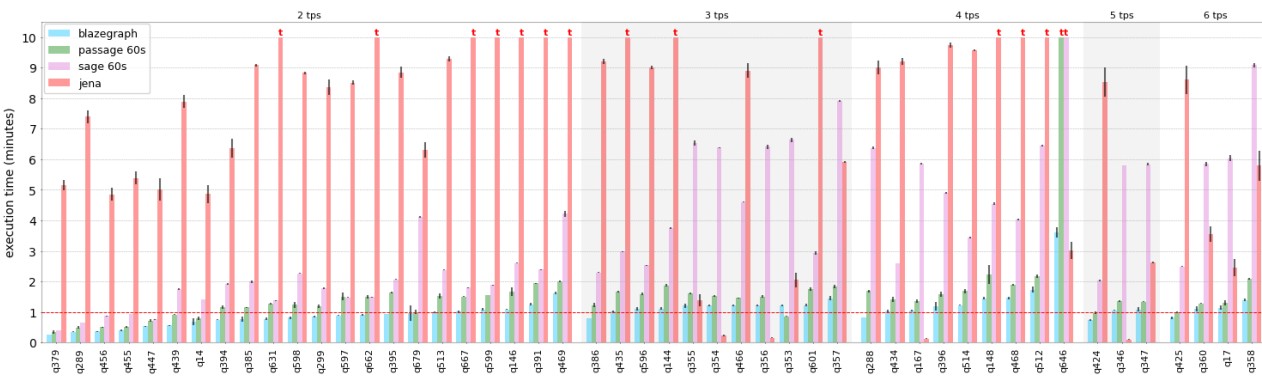

**Figure 5: Execution time comparison for BGPs queries group by number of triple patterns in the query with 4vCPU**

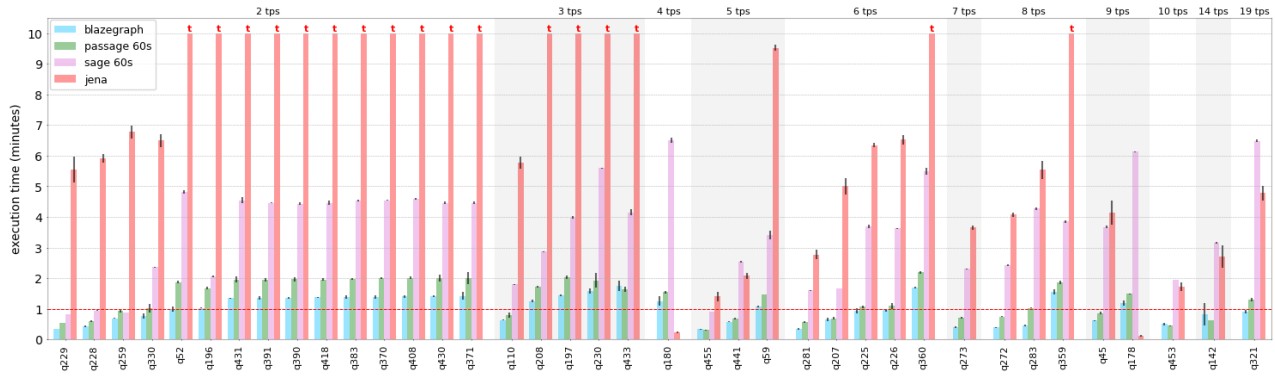

**Figure 6: Execution time comparison for OPTs queries group by number of triple patterns in the query with 4vCPU**

margin compare to Jena and SaGe-60s. All other engines do not impement intra-query parallelism and their performances are quite similar to whose observed in figures 3 and 4. It is worth noting that PASSAGE-60s's performance is slightly better in this configuration, while Jena and SaGe-60s do not get any improvement. Overall,

PASSAGE's performance without intra-query parallelism support remains in the same of order of magnitude than Blazegraph, and it leaves behind Jena and SaGe-60s in both configurations.

