# OpenReview forum: "Passage: Ensuring Completeness and Responsiveness of Public SPARQL Endpoints with SPARQL Continuation Queries"
_ACM.org/TheWebConf/2025/Conference — WWW 2025 Oral_

### Official Review · Reviewer_3jZ7 · 2024-11-29

**Novelty:** 5
**Technical Quality:** 6

**Review:**

This paper presents a novel approach to executing SPARQL queries over public endpoints while ensuring their completeness. This tackles a critical issue, as many real-world projects and use cases depend on public endpoints for functionality. The proposed solution is both simple and elegant (simple is good IMHO) and highly relevant to the conference audience.

The explanation is clear and supported by a formalization of the problem and illustrative examples, which make the solution easy to understand. The authors provide a thorough experimental evaluation, comparing two well-known SPARQL engines (Apache Jena and Blazegraph) with their solution and a prior approach addressing the same problem. I understood the selection of the later ones, but why other SPARQL engines are not included (GraphDB, Virtuoso, Oxigraph?). The results demonstrate the soundness of the solution and its effectiveness on systems with limited hardware resources (e.g., a single CPU). However, the approach shows limitations in comparison with Blazegraph when the number of CPUs increases.  The research questions of the experimental study are quite simple.

For the experimental evaluation, I noticed the absence of a configuration that considers the number of clients, which is a common factor in scenarios involving public endpoints. Public SPARQL endpoints typically handle multiple concurrent clients, and understanding the impact of this factor is essential. Specifically, I am curious about how the complexity of rewritten queries for continuous evaluation affects performance under varying levels of client concurrency.

Additionally, the discussion of the results primarily focuses on presenting numerical outcomes without delving into a deeper analysis. I was hoping for more detailed insights into the implications of the findings. Furthermore, specific measurements like those proposed in [1] for evaluating the continuous efficiency of query processing could significantly enhance the evaluation. Including such metrics would allow for a more nuanced and fine-grained comparison between the proposed solution and existing engines.

[1] Acosta, M., Vidal, M. E., & Sure-Vetter, Y. (2017). Diefficiency metrics: measuring the continuous efficiency of query processing approaches. In The Semantic Web–ISWC 2017: 16th International Semantic Web Conference, Vienna, Austria, October 21-25, 2017, Proceedings, Part II 16 (pp. 3-19). Springer International Publishing.

**Questions:**

1. **Table 2**: The meaning of `#TO/Total` is unclear. Could you clarify what this metric represents and its relevance in the evaluation context?

2. **Continuous Evaluation Metrics**: Why were continuous evaluation metrics, such as those suggested in [1], not included in the analysis? Including them could provide a more detailed and precise comparison of the performance.

3. **Complexity of Rewritten Queries**: What is the computational complexity of the rewritten queries, and how do they impact the execution time?

4. **Impact of Clients**: Why were clients not considered in the evaluation of the proposal? It is common for public endpoints to serve multiple clients simultaneously, and their impact on performance should be addressed.

5. **Source Code for Reproducibility**: Where is the source code to ensure the reproducibility of the results presented in the paper?

6. **Other SPARQL Engines**: Why were other SPARQL engines not included in the evaluation? Including additional engines could provide a broader perspective on the solution's performance.

**Reviewer Confidence:**

4: The reviewer is certain that the evaluation is correct and very familiar with the relevant literature

**Scope:**

4: The work is relevant to the Web and to the track, and is of broad interest to the community

---

### Official Review · Reviewer_eXJy · 2024-11-29

**Novelty:** 6
**Technical Quality:** 7

**Review:**

The paper introduces Passage, a method to ensure completeness of SPARQL queries that would timeout, by returning partial results and continuation queries.

The paper has several strength:
* The authors gave the right attention to formally detail the work and the several solution proposed, not forgetting to add examples to better understand the formal description
* The problem is well motivated and the content of the paper is clear since the beginning, allowing an easy reading
* The authors provide a (anonymised for the double anonymous review) repository with the used code for the experiments
* There is extra proof material in the appendix
* The evaluation is solid. I particularly appreciate the comparison between the versions with and without a 60s timeout (for both passage and sage), that helps to understand the impact of the continuation part. Moreover the authors justified well the choice of which competitors to have for the comparison of performances
* The implementation of the strategy using SPARQL features (BIND, UNION, ...) is smart and works

As weak point, the paper leaves some question unsolved (see questions)

**Questions:**

* Why do you tackle only NOT EXISTS filters? FILTERS with graph patterns may be even easier to evaluate. What is the rationale?
* How do you understand if a query is terminated? I try to explain. If we have 4 bindings in our query solution, imagining that we get the first 2, then timeout and continuation query #1, the other 2, then timeout. How do the system knows that this is finished? Or we simply have a continuation query #2, which will return  0 bindings and 0 continuation queries?

**Reviewer Confidence:**

3: The reviewer is confident but not certain that the evaluation is correct

**Scope:**

3: The work is somewhat relevant to the Web and to the track, and is of narrow interest to a sub-community

---

### Official Review · Reviewer_vaEK · 2024-11-30

**Novelty:** 5
**Technical Quality:** 6

**Review:**

This paper presents a continuation-based solution to the problem of endpoints delivering incomplete results due to size of answers.
The paper is elegantly written, presenting a conceptual model, theoretical analysis and experimental tests.
The approach taken (a recursive definition of continuation based on substructures) seems to be theoretically correct.
The experiments show (surprisingly) interesting results.

I have several question regarding the continuation queries returned.

**Questions:**

1- Seems that there is an assumption about the deterministic behaviour of the OFFSET feature in SPARQL. Did you test this assuming there is a server that process different queries in between (thus cache and order of evaluation could change?
2- Any hint about how would you deal with aggregation functions (that the conclusion indicate is for future work?) I ask this because it is crucial to be able to extend the method and my guess is taht your current approach does not extend easily to aggregation.
3- Can you tell us something about optimization (e.g. change of order of evaluation of triple patterns, etc.)

**Reviewer Confidence:**

3: The reviewer is confident but not certain that the evaluation is correct

**Scope:**

4: The work is relevant to the Web and to the track, and is of broad interest to the community

---

### Official Review · Reviewer_ZT1m · 2024-12-02

**Novelty:** 5
**Technical Quality:** 5

**Review:**

This paper proposes the creation of SPARQL queries that continue from a previous query that could not be completed. It formally describes the behavior of these Continuous SPARQL queries and proposes and evaluates an implementation on a SPARQL Endpoint.

# Strengths

- Relevance: The problem of obtaining no results due to timeout is common when exploring SPARQL Endpoints of large knowledge graphs.
- Utility: It logically aligns with the SPARQL standard, enabling its use from any SPARQL client.
- Reproducibility: The source code is available.

# Weaknesses

- Scope: It assumes that SPARQL queries always return a list of results, leaving aggregation operations out of the scope of this work. This is mentioned in section 3.1, but I believe it should be made more explicit in the paper.
- Inconsistency: The temporal dimension is not considered in the formalization of continuous queries, which could introduce unexpected partial results if new insertions occur in the knowledge graph during the retrieval process of several continuous queries. I believe a timestamp should be incorporated into the query so that partial results are consistent according to this time limit.
- Evaluation: The research questions evaluated are solely focused on execution time and the size and number of queries, but the quality of the obtained responses (completeness, exhaustiveness, uniqueness, etc.) is not evaluated.

**Questions:**

# Questions

- How are interruptions captured so that the partial response can be accumulated? What types of interruptions are handled? (only timeout?)
- What is the difference between PASSAGES (without quota) and Blazegraph or Jena? Why are their execution times so different if continuous queries should not be intervening here? (table 1)

**Reviewer Confidence:**

2: The reviewer is willing to defend the evaluation, but it is likely that the reviewer did not understand parts of the paper

**Scope:**

4: The work is relevant to the Web and to the track, and is of broad interest to the community

---

### Official Review · Reviewer_x6Sg · 2024-12-03

**Novelty:** 6
**Technical Quality:** 6

**Review:**

Approaches to ensure completeness and responsiveness for queries on linked data/rdf data on the web have been proposed but they are not compliant with SPARQL endpoints. Often, queries are interrupted due to fair use policies (such as time limits), leading to incomplete results.

The authors propose SPARQL continuation queries that, when reaching a time quotum, returns partial results along with a sparql query designed to return missing results (i.e., a continuation query).

They do this by returning, with the partial results, the same query with the OFFSET keyword is returned together with the first intermediate results. The returned query can then be evaluated again to include results that are not part of the intermediate results, ensuring completeness and responsiveness.


The contributions of the paper are:
* introduction of the SPARQL continuation queries
* PASSAGE (solution for the continuation problem) → it is not immediately clear what this is, is this a new engine/endpoint?
* query engine built on the blazegraph storage system
* evaluation on Wikidata benchmark

Comments:
+ I really appreciate the extensive examples given

+ the paper is written generally very clearly, although some concepts should be defined/written out explicitly (see comment below)

+ I appreciate the extensive experimental results, as well as the clear research questions to organise the results

+ I really like the simplicity of the idea and relevance of the work. Given that it is compliant with SPARQL endpoints, I think the impact of the work could be very good.

+ the code repository is very extensively documented, and there is even a live working demo


_-_ it is a bit clear in the list of contributions what PASSAGE exactly is (later I read it is a query engine), and how it compares to the SPARQL query engine developed on top of the Blazegraph storage.

_-_ some symbols/functions are not introduced. From the context it is easy to understand what they mean, but defining them explicitly would improve readability. For instance, card() → cardinality

In general, I really liked reading this paper: the proofs appear sound, the examples help with understanding, the evaluations look extensive, and the code & demo help understanding and uptake, and the work is very relevant to the semantic web community.

**Questions:**

* Why do you not compare passage and sage without the timeout? From table 1 it appears as if passage without timeout is also quicker in execution than the other query engines, why do you not include these in graph 3 and 4?
* Where do you think the big execution time difference comes from between for instance passage and Blazegraph for 1vCPU, without timeouts?
* you mention the join order caused a timeout, and fixing the join order fixed the issue. Is this issue now fixed, or does the correct join order depend on the query?

**Reviewer Confidence:**

2: The reviewer is willing to defend the evaluation, but it is likely that the reviewer did not understand parts of the paper

**Scope:**

4: The work is relevant to the Web and to the track, and is of broad interest to the community